# Polysaccharide-Based Supramolecular Hydrogel Coatings with Corrosion Barrier Zone for Protection of Patina Bronze

**DOI:** 10.3390/polym15163357

**Published:** 2023-08-10

**Authors:** Jiamei Zhang, Xia Huang, Jiachang Chen, Sheng Zhou, Junying Chen

**Affiliations:** 1School of Materials Science and Engineering, Zhengzhou University, Zhengzhou 450001, China; zhangjiamei97@163.com (J.Z.); zhousheng729@163.com (S.Z.); 2Henan Provincial Institute of Cultural Relics and Archaeology, Zhengzhou 450000, China; 3School of Chemistry, Zhengzhou University, Zhengzhou 450001, China; chenjy@zzu.edu.cn

**Keywords:** nanoclay, anticorrosion, gel coating, self-healing, buffer capacity

## Abstract

Protective coatings for bronze relics should adhere to the basic principles of cultural relic preservation, such as not altering the color and appearance of the artifacts, and being moderately combined with the artifacts to resist erosion due to external environments (such as water and gas). This paper presents the development of a physically crosslinked supramolecular hydrogel produced from guanidinium-based chitosan (GC). The hydrogel exhibits the excellent adsorption protection of bronze, and the addition of clay enhances the water barrier properties of the chitosan film. The supramolecular interaction between sodium polyacrylate/GC/clay confers corrosion buffering capability to the hydrogel coating in corrosive environments, and the gel coating can be self-healing at room temperature for 24 h. The fabricated nanocomposites were comprehensively characterized using various methods (Fourier transform infrared spectroscopy, X-ray diffraction, thermogravimetric analysis, X-ray photoelectron spectroscopy, scanning electron microscopy, etc.). The electrochemical properties of coated specimens were evaluated, and the impedance spectrum revealed a large impedance arc indicating high charge resistance, which has a corrosion resistance effect.

## 1. Introduction

Bronze artifacts are an important heritage of ancient Chinese civilization. Bronze artifacts that have been excavated and stored are subject to corrosion because of their burial environment and inherent characteristics. This corrosion can cause the loss of surface inscriptions and ornaments as well as ulceration and perforation, leading to irreversible damage. The prevention of continued rusting and deterioration of bronzes in collected artifacts has been an urgent worldwide problem in the field of cultural conservation [1,2]. The application of a protective layer on bronze can effectively protect it by inhibiting and slowing down further corrosion aggravation caused by environmental changes. In light of the basic principle of cultural relic protection and restoration, transparent and flexible gel protective coatings have received widespread attention [3,4]. Nevertheless, the barrier performance of coatings tends to be weakened over time due to complex environmental conditions or corrosion attacks, resulting in subsequent damage. Furthermore, it is crucial to ensure that the self-healing capability remains effective to prevent any additional corrosion damage until the necessary repairs can be carried out. Thus, it is imperative to develop novel coatings with the dual abilities of anti-corrosion and self-healing.

Chitosan is a linear bio-polymer derived from chitin which is found in the shells of crustaceans [5]. It has bio-adhesive and antibacterial [6] effects and acts as a chelating agent [7] and antioxidant. Chitosan and its derivatives are widely used as corrosion inhibitors due to the presence of electron-rich sites. Amino (–NH_2_) and hydroxyl (–OH) groups serve as adsorption sites and can protect against corrosion attacks [8,9], enabling chitosan to bind to metal surfaces [10]. However, chitosan is highly soluble in acidic aqueous media and insoluble in alkaline solutions [11]. To improve the water solubility of chitosan, researchers studied guanidinylated chitosan [12]. They found that the guanidine group enhanced the antimicrobial activity of chitosan under neutral conditions. The chemically modified guanidinium-based chitosan (GC) exhibited superior abilities in metal binding and anticorrosion compared with unmodified chitosan [10]. Additionally, chitosan has excellent film-forming properties [13]. Although the chitosan coatings have good air tightness, their ability to block water is weak [14]. Furthermore, the mechanical properties of single-component chitosan coatings are poor. By incorporating nanoclay, the coating maintains its original level of airtightness while improving its resistance to moisture. Additionally, the inclusion of nanoclay allows the coating to self-heal to some extent. Nasim et al. [15] developed nanohybrid interpenetrating polymer network (IPN) hydrogels of polyvinyl alcohol (PVA)-alginate containing lithium saponite (LN). It has been shown that the addition of LN nanosheets to IPN hydrogels of PVA-alginate leads to nanohybrid hydrogels with appropriate mechanical, physical, and biological properties. The study of polymer/clay nanocomposites was first reported in 1961. At that time, Blumstein [16] demonstrated the polymerization reaction of vinyl monomers embedded in montmorillonite (MMT).

LN is a synthetic smectite clay mineral of the trioctahedral 2:1-type layered silicates [17]. Compared with other silicate layers, LN exhibits superior swelling performance, a larger specific surface area, and greater cation exchange capacity. Organic cations can replace Na^+^ and K^+^ in LN. Additionally, because of the ion exchange mechanism, an intercalation structure can be realized. Although LN can adsorb bacteria, it does not have an antibacterial effect. Therefore, an organic material is usually inserted into the lithium soapstone flake. The introduction of guanidinium groups enhances the surface charge of chitosan [18], enabling better formation of crosslinked networks with LN and facilitating bonding with the bronze surface.

Numerous studies have documented the effectiveness of chitosan and its derivatives combined with diverse organic and inorganic substances as corrosion inhibitors for safeguarding bronze. In this paper, nanoclay particles were first mixed in an aqueous solution, and swelling occurred when they were dispersed in water, gradually cleaving into discrete disc-shaped particles. The resulting aqueous phase clay suspension consisting of exfoliated clay particles was homogeneous and transparent, because the clay particles have a negatively charged surface and a positively charged edge [19]. When the dispersant sodium polyacrylate (PAANa) was added to the suspension, it adsorbed onto the positively charged edges of the clay particles to produce electrostatic interactions and disperse the nanoclay. In addition, the subsequent addition of GC created electrostatic interactions and hydrogen bonding interactions with the LN surface to form a crosslinked network [20,21,22]. The surface mechanism for bronze is shown in Figure 1. The experimental results show that the supramolecular gel (LN/PAANa/GC) with guanidine chitosan had better corrosion resistance than the gel coating (LN/PAANa) without guanidine chitosan. In addition, the gel coating has a corrosion barrier zone and good self-healing ability.

## 2. Experimental

### 2.1. Materials

Ethylene glycol chitosan (60%), PAANa (poly (acrylic acid sodium salt)), 1h-pyrazole-1-carboxamidine hydrochloride (98%), sodium magnesium lithium silicate (lithium saponite), sodium chloride (NaCl), Acetone, hydrochloric acid (HCl, 36–38%) were purchased from Shanghai Aladdin Technology Co., Ltd. (Shanghai, China). *N*,*N*-diisopropylethylamine (99%) was supplied by the Tianjin Comio Chemical Reagent Co. (Tianjin, China). Constant temperature magnetic stirrer (85–2) was supplied by the Jiangsu Zhongda Instrument Technology Co. (Wuxi, China).

### 2.2. Guanidinylation Modification of Ethylene Glycol Chitosan

Ethylene glycol chitosan (1.0 g) and 1h-pyrazole-1-carboxamidine hydrochloride (3.85 g) were added to a beaker and dissolved in 20 mL of deionized water. Then, 4.34 mL of *N*,*N*-diisopropylethylamine was added and the reaction was conducted at room temperature for 24 h. The free 1h-pyrazole-1-carboxamidine hydrochloride was then removed by dialysis with deionized water using a dialysis bag (MWCO 3500 da) for 24 h. The dialysis product was frozen in a 40 °C refrigerator for 90 min. Subsequently, the frozen samples were dried in a vacuum dryer (DZF-6020AB, Beijing Zhongxing Weiye Instrument Co., Beijing, China) for 24 h to obtain the GC samples.

### 2.3. Preparation of Supramolecular Hydrogels

LN (0.2 g) was added to 50 mL of deionized water. Magnetic stirring was performed at 1000 rpm for 30 min. Then, 0.1 g of PAANa was added to the well-dispersed LN solution. Magnetic stirring was performed at 500 rpm for 10 min. Finally, GC was added and stirred for 30 min. The hydrogel sample of LN/PAANa/GC was obtained after the mixing was completed.

### 2.4. Bronze Sheets Coating

The obtained gel was applied to the surface of the bronze sheet (5 × 3). After drying at 35 °C for 12 h, thin sections were prepared for subsequent characterization.

### 2.5. Evaluation of Self-Healing Performance of the Coatings

Cracks were made on the gel coatings by a scalpel blade. The scratched gel coating was kept at ambient temperature for 24 h, and then its self-healing ability was observed.

### 2.6. Artificial Aging

To evaluate the long-term protective effect of the gel coating, we conducted artificial aging of the coated copper sheets. HCl vapor was prepared by heating 1.0 M HCl in water in a closed glass vessel at 50 °C and 100% relative humidity (RH) (constant temperature and humidity test chamber). Then, the bronze sheet was subjected to an accelerated corrosion test for 60 h. Electrochemical analysis was performed on the bronze sheet before and after corrosion. The changes in the protective effects of the gel coatings were detected using electrochemical analysis.

### 2.7. Characterization

Fourier transform infrared (FTIR) spectrometry was performed using a Nicolet IS10. A comparison of the IR spectra of glycol chitosan and GC was obtained between 4000–5000 cm^−1^ using the potassium bromide pellet method. Patina bronze flakes coated with the gels without guanidine chitosan or the composite gels were measured in total reflection mode.

Thermogravimetry/derivative thermogravimetry (TG/DTG) was heated from room temperature to 600 °C using a NETZSCH STA 449 F3/F5 under nitrogen flow at a ramp rate of 10 °C/min.

Gel coatings were analyzed using an X-ray diffractometer (D8 ADVANCE, Bruker, Germany). The tube current was 40 mA and the Cu target was used to radiate the sample.

Synchrotron radiation small-angle X-ray scattering (SAXS) experiments were performed on a Xeuss 2.0 of Sanop (France) with an X-ray wavelength of 1.54189 Å. The two-dimensional (2D) SAXS data was converted to one-dimensional (1D) intensity (*I*(*q*)) as a function of the scattering vector *q* [*q* = 4*π**sinθ*/*λ*] by circular averaging, where 2*q* is the scattering angle. SAXS was used for the analysis of the blank gels (chitosan-free gel).

X-ray photoelectron spectroscopy (250Xi, Thermo Fisher Scientific, Waltham, MA, USA) was used to evaluate the binding of the gel to the patina bronze sheet. The binding energy reference C1s peak was 284.80 eV.

A spectrophotometer (CS-5960GX) was used to analyze the patina bronze flakes coated with hydrogel and the uncoated patina bronze flakes.

A field emission scanning electron microscope (SU8020, HITACHI, Tokyo, Japan) was used to observe the surface morphologies of the gel coating and bronze binding sites, as well as the gel coating on the bronze surface. The working voltage was 3 kV, and the working mode was the secondary electron mode [23].

A traditional three-electrode system comprising the reference (saturated calomel), working, and auxiliary (platinum) electrodes was used in the experiment to monitor the corrosion potential of the studied electrode (Wuhan Crest Instrument Co., Wuhan, China). The frequency range was 0.1–105 Hz, and the AC amplitude was 30 mV. The measurements are represented as Nyquist plots or Bode plots.

To evaluate the long-term protective effect of the gel coatings, the coated and uncoated bronze sheets were heated in closed glass containers at 50 °C in the presence of HCL vapor (1.0 M HCl in water) and 100% RH. The sheets were then subjected to the 60 h accelerated corrosion tests, and the electrochemical analysis of the bronze sheets was performed before and after corrosion.

## 3. Results and Discussion

### 3.1. Chemical Structure and Thermal Analysis of Gel Coating

FTIR spectroscopy was used to confirm the formation of the GC and hydrogel networks. Figure 1a shows that the ethylene glycol chitosan sample exhibits characteristic absorption bands [11,24,25] at 3317 cm^−1^ (–OH stretching), 2928 cm^−1^ and 2870 cm^−1^ (C–H stretching), 1359 cm^−1^ (C–N stretching), 1456 cm^−1^ (N–H bending), and 1649 cm^−1^ (C=O stretching), which are related to the residual N−acetyl group. The peaks observed 1655 cm^−1^ and 1359 cm^−1^ are related to GC. The comparison of the spectra of these two samples [26] indicate that strong absorptions occur at 1655 cm^−1^ and 1359 cm^−1^ in the spectrum of the GC, which appeared blue-shifted, indicating the successful introduction of the guanidinium moiety.

As shown in Figure 1b, the characteristic peaks of GC are 1456 cm^−1^ (in-plane bending vibration absorption peak of secondary amine N–H) and 1363 cm^−1^ (stretching vibration of C–N). This demonstrates that the bands of N–H shifted to 996 cm^−1^ when LN and Pagana were added to the hydrogels in the composites, possibly resulting from hydrogen bonding between CS and LN or PAANa. The characteristic peak of C=O in the LN/PAANa hydrogel is observed at 1658 cm^−1^. Figure 1b shows that the GC has no obvious C=O characteristic peak. The characteristic C=O peak at 1660 cm^−1^ appears after the formation of the LN/PAANa/GC hydrogel, and the intensity of the absorption peak considerably decreases, indicating intermolecular interactions between LN and GC. For the LN/PAANa hydrogel, the peak observed at 646 cm^−1^ is a characteristic peak of Si–O. The electrostatic interaction between LN and GC caused a redshift of the characteristic peak in the LN/PAANa/GC hydrogel, which appeared at 613 cm^−1^. This indicates that a supramolecular gel network was formed in the hydrogel containing GC.

After LN/PAANa gel and LN/PAANa/GC coating are applied on the bronze plate, the copper on the surface of the bronze can react with the carbon–oxygen double bond in the carbonyl group, forming a metal–carbonyl bond. This type of bonding can provide strong adhesion, allowing the carbonyl functional group to firmly attach to the surface of bronze. Therefore, the characteristic peak of the coated gel layer C=O becomes weaker. In addition, the LN/PAANa/GC gel coating contains an –OH peak at 3200–3500 cm^−1^ and an N–H peak at 1456 cm^−1^. From Figure 1c, it can be seen that the characteristic peaks of –OH and N–H are weakened after coating. In conclusion, the analysis reveals the adsorption between the coating and bronze.

Thermogravimetric analysis (TGA) of the hydrogels was performed to evaluate the thermal stability of the samples (Figure 2). As shown in the corresponding TGA curve, the weight of all samples gradually decreased with the increasing temperature. The initial weight loss of the hydrogels at ~100 °C was due to the loss of water. The LN/PAANa gel showed a weight loss of ~20% at 450 °C, indicating that the interaction between PAA and clay was electrostatic and that ion exchange did not occur. Moreover, the thermal stability decreased after adding GC because the interaction between GC and the nanoclay included both ion exchange and intermolecular interactions such as electrostatic adsorption. The ion exchange interaction led to an increase in GC in the nanoclay interlayer. Therefore, the LN/PAANa/GC gel showed greater weight loss. Furthermore, because electrostatic interactions are more stable than ion exchange interactions at high temperatures, the decomposition temperature of the LN/PAANa gel was slightly higher than that of the LN/PAANa/GC gel.

### 3.2. XRD

XRD patterns were used to evaluate the interaction of the nanoclay with GC and PAANa, as well as the gels with a patina bronze sheet. XRD was performed on the blank gel samples (LN/PAANa and LN/PAANa/GC), LN, and patina bronze sheets coated with the LN/PAANa/GC hydrogel (Figure 3). For the LN/PAANa gel, multiple sharp diffraction peaks (2*θ* = 19.8,28.5, 34.9) were displayed, which corresponded to the different crystal faces ((100) (211) and (300), respectively) of LN [27]. This indicated that LN and polyacrylic acid were only electrostatically adsorbed, no intercalation or exfoliation occurred, and the crystal structure of LN was not considerably affected. Because GC is a positively charged polyelectrolyte and the LN surface is negatively charged, the intercalation of the polysaccharide is mainly governed by an ion exchange mechanism. With the introduction of GC to the LN/PAANa system (Figure 3a), GC underwent ion exchange with LN. In addition, the original lattice structure was destroyed because of the large degree of LN exfoliation. Therefore, in the LN/PAANa/GC gel, the original diffraction peak of LN located between 20–70° was not observed. This indicates that lithium saponite is well dispersed in the LN/PAANa/GC gel matrix. The 79–83° conversion corresponded to a crystal plane spacing range of 0.7769 to 0.8049 Å, corresponding to the (200) crystal plane of LN. This crystalline surface did not appear in the LN/PAANa gel, indicating that the binding state of LN changed after the introduction of GC. The appearance of diffraction peaks at small angles indicated an increase in the layer spacing of LN.

As shown in Figure 3b, compared with the blank gel, several new sets of diffraction peaks appeared after the LN/PAANa/GC hydrogel was coated on the bronze sheet. This occurred because the electrostatic adsorption and ion exchange with LN weakened after GC was adsorbed on the bronze surface; thus, the LN diffraction peak reappeared. Additionally, the adsorption of GC chain segments to the bronze surface limited chain movement, which was consistent with the TGA results. It was also shown that GC in the LN/PAANa/GC gels tended to adsorb more on the bronze surface when interacting with LN. After the “unbinding” of GC and LN, the interaction between LN and PAANa formed a double layer of protection for the bronze flakes. Furthermore, after aging, the distribution of the diffraction peaks of the gel coating at 42° was more consistent. This indicated that the composite layer of LN and PAANa consumed the hydrogen ions in the acid mist through ion exchange of sodium ions with hydrogen ions, which led to a more uniform thickness distribution of the LN wafers. This was verified in the energy dispersive spectroscopy (EDS) analysis of the gel coating after corrosion.

### 3.3. SAXS

The 2D SAXS data of the samples are shown in Figure 4. This study was a four-component system consisting of nanoclay, PAANa, GC, and water, and the entire X-ray scattering intensity can be described by the sum of several partial scattering functions. The X-ray scattering will be dominated by the signal from clay particles because the electron density of clay in an aqueous solution is much larger than that of GC units. Δ*ρ*, *P*(*q*), and *Sexp* (*q*) are the scattering length density difference between the clay particles and matrix, the shape factor of the clay particles, and the experimental structure factor, respectively. *K* is the experimental constant and nclay is the number density of clay particles with volume vclay [21].

The 2D SAXS data are converted to a 1D intensity *I*(*q*) as a function of the scattering vector *q*.
(1)$$I\left(q\right)=Kn_{clay}V_{clay}^{2}{∆\rho }^{2}P(q)S_{exp}(q)$$

The 2D SAXS data are converted to a 1D intensity *I*(*q*) as a function of the scattering vector *q*.
(2)q=4πsinθ / λ
where *θ* and *λ* are the scattering angle and wavelength of the X-rays, respectively.
(3)$$P\left(q\right)=4\int_{0}^{\pi /2}\left[\frac{{sin}^{2}(qHcos\beta )}{{\left(qHcos\beta \right)}^{2}}\right][\frac{J_{1}^{2}(qRsin\beta )}{{\left(qRsin\beta \right)}^{2}}]sin\beta d\beta$$

Here, 2*H* and *R* are the thickness and radius of the disk-shaped particle, and *β* is the angle between the scattering vector *q* and the axis of the disk-shaped particle. *J*_1_ denotes the first-order Bessel function, since the inhomogeneity of the radius of the disk ions and the inhomogeneity of the Gaussian distribution are taken into account (*R* = 13 nm, 2*H* = 1.0 nm).

All samples showed clear scattering patterns, indicating the presence of multiphase microstructures. The 2D scattering data showed isotropic scattering rings (circles), indicating the isotropic multiphase structure of the gel. With *q*_max_ = 0.14 and *L* = 2*π*/*q* = 45 nm, the nanoclay diameter was ~30 nm, indicating that some nanoclays are in direct contact with each other. The peak height decreased in the absence of GC, probably due to the presence of exfoliated nanoclay structures. After the introduction of GC, the LN binding state changed and the layer spacing increased. From the XRD pattern (Figure 3a), it can be concluded that the layer spacing is widened to 16.6314 Å.

### 3.4. X-ray Photoelectron Spectroscopy

The patina bronze flakes coated with LN/PAANa/GC and LN/PAANa gels were characterized using XPS (X-ray photoelectron spectroscopy). The XPS spectrum of the LN/PAANa/GC gel coating (Figure 5a) showed three peaks at 284.80, 399.93, and 532.17 eV corresponding to C1s, N1s, and O1s, respectively. The high-resolution C1s (Figure 5b) showed three peaks at 284.80, 286.20, and 288.59 eV for the patina bronze flakes coated with the LN/PAANa/GC gel, corresponding to C–C, C–O/C–N, and C=O, respectively [28,29]. The presence of C–N was not detected in the patina bronze flakes coated with the LN/PAANa gel. The N1s peak (Figure 5c) was more obvious in the rusting bronze sheet coated with the LN/PAANa/GC gel, whereas the N1s appeared as two peaks for N–H. This indicated that GC was successfully intercalated into the interlayer of LN to form a layered structure. The peaks in the N1s spectrum were slightly shifted in the direction of high binding energy, demonstrating the presence of hydrogen bonds. Analysis of the high-resolution O1s spectrum of the same flakes coated with the LN/PAANa/GC gel (Figure 5d) revealed three distinct peaks attributed to the C–O peak at 531.34 eV, C–O–C present in guanidine chitosan, C–OH and Si–O in LN at 532.17 eV, and C=O at 533.01 eV, respectively [30]. The xps results also showed the successful synthesis of gel coatings.

### 3.5. Colorimetric Difference

The colorimetric parameter data (*L**, *a**, *b**) and color change (*E*) were calculated using the color of the uncoated sample as a reference. In the analysis of the patina bronze flakes coated with the LN/PAANa/GC hydrogel and bare bronze flakes (Table 1), it was greater than 0, indicating a color change.
(4)$$\Delta E=\sqrt{{\left(∆L*\right)}^{2}+{\left(∆a*\right)}^{2}+{\left(∆b*\right)}^{2}}=7.638\;$$

The decrease in the *L** values reflected a color shift to dark green, which is consistent with the observations reported in the literature [31].

### 3.6. Micromorphology of Gel Coating on the Bronze Surface

The micromorphology of the surface of the gel-coated patina bronze flakes was analyzed using scanning electron microscopy (SEM). Figure 6 shows the patina bronze flakes coated with the LN/PAANa/GC gel and those coated with the LN/PAANa gel. As shown in Figure 6a,c, all coated gels showed undulations, indicating that the gels formed a tight bond with the bronze sheet. From the analysis of Figure 6b, irregular particles precipitated from the surface of the patina copper bronze sheet coated with the LN/PAANa/GC gel. The scale varied from 20 to 50 μm.

Elemental quantification of the gel-coated bronze sheets was performed based on the EDS analyses. The results are shown in Figure 7, which clearly shows the distribution of C, O, Mg, and Si elements in the LN/PAANa/GC gel composite coating. The main body of the hydrogel consisted of PAANa, LN, and the GC structure. Furthermore, lithium has a low atomic number and its X-ray emission is relatively weak compared to other elements. The low X-ray intensity makes it difficult to accurately detect and quantify the lithium content using EDS. Figure 7 shows that the Mg and Si elements in the gel coating were uniformly distributed in high concentrations, confirming that the analyzed material was LN. The LN/PAANa/GC gel had higher concentrations of Mg, Si, and Na elements compared with the LN/PAANa hydrogel. This was because the binding state of LN changed, the layer spacing increased, and the exfoliated structure appeared. As shown in Figure 7, the elemental N content on the surface was low, indicating that GC was more likely to adsorb on the bronze surface when interacting with LN.

Figure 8 shows the SEM images of the cross-section of the rusting copper bronze sheet coated with the LN/PAANa/GC composite gel. As shown in Figure 8a, the thickness of the gel coating was ~4 μm. The cross-sectional electron microscopy images showed that LN was dispersed in the hydrogel in an exfoliated structure. Figure 8 and Figure 9 also show that the hydrogel bonded well with the surface of the patina bronze sheet. Figure 9 shows the high N content on the bronze sheet surface, which was consistent with the surface electron microscopy EDS results (Figure 7). It also indicated that GC was more likely to adsorb on the bronze surface. Furthermore, the uniform distribution of Si, Na, and Mg elements indicated that LN was uniformly distributed inside the gel coating.

### 3.7. Self-Healing Performance of the Gel Coating

The self-healing properties of LN/PAANa/GC gel coatings were investigated. Figure 10a is dried directly after coating on the bronze sheet, and Figure 10b is dried after 24 h at room temperature. As shown in Figure 10, the gel coating is completely self-healing. After the self-healing effect of the gel coating, it still maintains a tight bond with the bronze plate. The presence of hydrogen bonds and electrostatic interactions within the gel. These supramolecular interactions provide the gel coating with an effective self-healing ability.

### 3.8. Characterization of Gel Coating Electrochemical Properties

Figure 11 shows the Nyquist, Bode, and Bode-Phase plots of bare patina bronze sheet electrodes immersed in 3.5% NaCl solution and patina bronze sheet electrodes coated with gels. In the case of immersed bare samples, the equivalent circuit consisted of a solution resistor (R_s_) and a parallel constant phase element (CPE_dl_) (Figure 12a). After coating the bronze sheet, additional parallel resistors (R_f)_ and constant phase elements (CPE_f_) need to be considered, where R_c_, R_f_, and R_ct_ represent the solution resistance, gel coating resistance, and charge transfer resistance, respectively. CPEf is the constant phase element associated with the membrane, and CPE_dl_ is the constant phase element between the electrode surface and the corrosion medium (Figure 12b) [32].

By comparing the impedance values, the corrosion resistance of uncovered and organic coating-covered electrodes can be quantitatively evaluated. The gel coating is related to the magnitude of the impedance arc; the larger the impedance arc, the better the corrosion resistance [33]. As shown in Figure 11, an impedance value of 102 Ω was observed in the low-frequency region, indicating that the bare bronze could be corroded. Conversely, the bronze sheet with the gel coating displayed high impedance values in the low-frequency impedance region, the impedance arc of the coated electrode increased, and the maximum value of the phase angle obtained was considerably higher. In Figure 11, the impedance arc of the LN/PAANa/GC gel-coated electrode was larger than that of the LN/PAANa gel-coated electrode, showing a higher |Z| and a phase angle of nearly 30° at high frequencies, indicating that the gel coating had good capacitive properties.

The fitting circuit shown in Figure 12b was selected for fitting according to the gel coating. The fitting results showed that the total impedance values of the LN/PAANa/GC gel electrode and the LN/PAANa gel electrode were 6250 Ω and 4000 Ω, respectively. The results showed that the LN/PAANa/GC gel coating had a superior impedance effect; therefore, the impedance effect increased with the addition of guanidine chitosan. From the SEM, it was also clear that the composite gel containing guanidine chitosan was more resistant to corrosion.

### 3.9. Artificial Aging

To evaluate the long-term protective effect of the gel coating, we conducted artificial aging of the coated copper sheets. HCl vapor was produced by heating 1.0 M HCl in water in a closed glass vessel at 50 °C and 100% RH. The bronze sheet was subjected to an accelerated corrosion test for 60 h and electrochemical analysis was performed on the bronze sheet before and after corrosion. The changes in the protective effects of the gel coating were detected using electrochemical analysis. As shown in Figure 2, after corrosion, GC tended to adsorb on the surface of the bronze after unbinding with LN. The interaction between LN and PAANa formed a double layer of protection for the bronze flakes. The ion exchange between Na^+^ and H^+^ consumed the H^+^ in the acid mist, and Na^+^ formed NaCl with the Cl^−^ in the acid mist. As the salt precipitation consumes the Cl^−^ in the acid mist, it forms a hierarchical structure. This gives the gel coating a corrosion barrier zone.

The SEM images of the surfaces (after aging) of LN/PAANa/GC gel-coated and LN/PAANa gel-coated patina copper bronze sheets are shown in Figure 13. The surface of the LN/PAANa/GC gel coating (Figure 13a,b) was relatively smooth and maintained the flatness of the bronze sheet. The hydrogel remained on the surface of the bronze sheet. In contrast, the LN/PAANa gel coating (Figure 13c,d) was absent, and the bronze sheet showed obvious corrosion. This demonstrated that the LN/PAANa gel was not as effective against corrosion as the LN/PAANa/GC gel. After acid mist corrosion, due to ion exchange there is salt precipitation, which consumes Cl^−^. This allows the LN/PAANa/GC coating to have a corrosion barrier zone. Thus, the gel coating retains its integrity.

Figure 14 depicts the results of the EDS analysis performed on the bronze flakes subjected to aging. The LN/PAANa/GC gel coating showed an increase in Na. Furthermore, in conjunction with SEM (Figure 13b), the aging process resulted in salt precipitation with a size of ~1 μm. The XRD analysis (Figure 15) indicated the presence of NaCl, corresponding to 2θ = 75.302° within the spectrum for the (420) crystal plane. The surface of the LN/PAANa gel-coated rusting bronze sheets did not show NaCl precipitation because GC underwent ion exchange with LN but not with PAANa. The binding force on the sodium ion reduced after ion exchange. Additionally, surface EDS analysis (Figure 6) demonstrated that the LN/PAANa/GC gel coating showed an increase in Na^+^ concentration after the introduction of GC. Therefore, it was determined that the Na^+^ in NaCl was derived from LN.

The bronze sheet was subjected to an accelerated corrosion test for 60 h. HCl vapor was produced by heating 1.0 M HCl in water in a closed glass vessel at 50 °C and 100% RH. As shown in Figure 16, the fitting results showed that the gel coating resistive impedance arc increased compared with the uncoated bronze sheets, indicating the continuity of the corrosion inhibition properties. It also shows that the gel coating has corrosion buffering capacity. The total impedance value of the LN/PAANa/GC coated gel bronze sheet after aging was 5500 Ω, demonstrating that after aging for 60 h, the gel coating still had some corrosion resistance. Figure 17 shows the photographs taken by the camera. It shows images of LN/PAANa/GC gel-coated (Figure 17a) and LN/PAANa gel-coated (Figure 17b) patina copper surfaces (after aging) and images of LN/PAANa/GC gel-coated (unaged) (Figure 17c). Figure 17a,c, show that there was no considerable corrosion on the surface of the LN/PAANa/GC gel-coated bronze sheet, whereas there was considerable corrosion on the surface of the LN/PAANa gel-coated bronze sheet (Figure 17b). These results indicate that the LN/PAANa/GC gel had excellent anticorrosion properties.

## 4. Conclusions

In this study we combined LN, GC and PAANa to prepare a hydrogel coating with a sealing effect. After coating a bronze sheet, a fully coated and almost transparent protective gel coating was obtained. The experimental results clearly showed that the LN/PAANa/GC gel coating has good self-healing ability and provides corrosion resistance in very accelerated environments (acid mist). At the same time, the gel coating has a corrosion barrier zone because of the ion exchange. The excellent corrosion resistance of this gel was attributed to the –NH_2_ and –OH groups in GC, acting as adsorption centers that can provide protection from corrosion attacks. Furthermore, LN keeps the gel airtight and improves the moisture barrier properties. This work demonstrates the feasibility of a supramolecular hydrogel comprising a guanidine chitosan and nanoclay composite for bronze protection. These results have important implications for further research into the use of new composite materials for the protection of bronzes.

## Data Availability

The data presented in this study are available on request.

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
