# Peer review of "Polysaccharide-Based Supramolecular Hydrogel Coatings with Corrosion Barrier Zone for Protection of Patina Bronze"

_polymers, 2023, doi:10.3390/polym15163357_

Round 1

Reviewer 1 Report (Previous Reviewer 2)

Unfortunately, over the past three weeks, not all the comments were corrected by the authors correctly, and some only partially. We are forced to call all the same remarks major and offer to correct them necessarily or provide reasoned objections.

Point 1: The introduction should be redone. The introduction should be shortened, the purpose of the work should be clearly stated and remove your results. The scheme may be presented in the discussion of the paper or in the conclusions. It is reasonable to combine schemes 1 and 2.

Response 1: We feel great thanks for your professional review work on our article. I don't think it is necessary to shorten the introduction. Because, we need to draw out the necessity of adding nanoclay to form supramolecular gels from the poor mechanical properties and poor moisture barrier ability of single-component hydrogels. Also since the antimicrobial properties of single component chitosan are not as good as guanidinium based chitosan, the literature needs to be cited to indicate this (Second paragraph ). In addition, the purpose of the work is shown in the first paragraph of the introduction. It may not make sense to combine Scheme 1 and 2. Since Scheme 2 shows a hydrogel coating after accelerated corrosion with structural changes, Scheme 2 is more reasonable to be placed in 3.9.

Comments on the response. The aims and specific tasks of the work should be formulated more clearly and are usually placed at the end of the Introduction (e.g., as in Rennert, M.; Hiller, B.T. Polymers 2023, 15, 2985 doi.org/10.3390/polym15142985). The scheme is not clear now and needs clarification. What is this multicolored object? Schemes are needed to facilitate the understanding of processes, unfortunately this is not the case now. I would like to understand more clearly what kind of colored object is above the scheme. The same is true for scheme 2.

Point 2: Schemes 1 and 2 can hardly be called "mechanisms", the mechanism requires bringing stages. This is most likely the principle. At the same time, great respect to the authors for the fact that they bring schemes. Unfortunately, in their present form they are difficult to understand. A more detailed decoding of everything with arrows that is present on them is required. You should also increase the font size.

Response 2: We think this is an excellent suggestion. I have presented the revised image in the manuscript.

Comments on the response. This remark has not been corrected, or not completely corrected.

Point 3: It is recommended to make a table with abbreviations and formulas of the substances used and their combinations.

Response 3: We think this is an excellent suggestion. I have presented the revised text in the manuscript. It is highlighted by blue in the revised version for easier tracking.

Comments on the response. This remark has not been corrected. The table is only partially filled in. The table should be located in a special part of the Appendix (which is in the template). What is it now on Line 472? No blue highlights were found in the text.

Point 4: FTIR spectra, TG and DTG curves reasonable represent in supplementary materials.

Response 4: Thank you for pointing this out. However, I think FTIR spectra, TG and DTG curves should be shown in 3.1 because IR spectra can prove the formation of guanidinium-based chitosan and supramolecular gel networks, which plays a crucial role and would be more appropriate to be placed in the first section. Simultaneous thermal analysis, which can verify the interaction between the supramolecular gel network. 2

Comments on the response. The new figure has all the same comments, in addition, the figure caption is not added.

Point 5: Line 95. What is formula of sodium chloride acetone?

Response 5: We were really sorry for our careless mistakes. “sodium chloride acetone” is a spelling mistake. The correct text should be “Sodium chloride (NaCl), Acetone”. It is highlighted by blue in the revised version for easier tracking.

Comments on the response. No blue highlights were found in the text.

Point 6: Section 2.1. Not all countries-manufacturers of this equipment are indicated in the description of the devices.

Response 6: We think this is an excellent suggestion. The lines 97-106 from this subsection were removed. The information for each instrument was given in section 2.7. Therefore all the countries and manufacturers of the equipment will appear cluttered in the text. I also found a lot of literature that didn't indicate the manufacturer or country of the device.

Comments on the response. In journals MDPI, such information should usually be.

Point 7: It is recommended to break at least into paragraphs according to the description of each method.

Response 7: Thank you for this suggestion. I don't think it is necessary to break the description into paragraphs. Breaking the description into paragraphs may lead to a long and complicated article.

Comments on the response. A slight remark, however, is not corrected. This is now Section 2.7 is difficult to read, all the methods are piled up. Perhaps the authors did not understand the question.

Point 8: What is the chemical composition investigated “bare” bronze and “rusted” bronze?

Response 8: Thank you for your question. Bare bronze is rusted bronze, bare bronze sheets are rusted bronze sheets without gel coat. In order to separate it from the gel-coated corroded bronze flakes. The main components of rusty bronze flakes are CuOCu2OCuCO3·2Cu(OH)2CuCl2·3Cu(OH)2CuCl.

Comments on the response. This explanation should be added in the text and possible to the table of abbreviations.

Point 9: Why authors used term rusted bronze and no patina bronze (https://doi.org/10.1016/j.culher.2021.09.007)? The term rusted often concern of iron.

Response 9: Thank you for your question. This is because of the rusting that occurs when bronze is corroded. The patina of bronze (alkaline copper chloride) is greenish powdery, loose and swollen. Because some paper will use patina bronzeas the expression. In fact, both formulations are fine.

Comments on the response. A huge amount of literature works with the term Patina . Authors are strongly recommended to decide to change the title of the article using this term. It will be much more understandable for a wide range of readers.

Point 9: Why authors used term rusted bronze and no patina bronze (https://doi.org/10.1016/j.culher.2021.09.007)? The term rusted often concern of iron.

Response 9: Thank you for your question. This is because of the rusting that occurs when bronze is corroded. The patina of bronze (alkaline copper chloride) is greenish powdery, loose and swollen. Because some paper will use patina bronzeas the expression. In fact, both formulations are fine.

Comments on the response. The answer is incorrect. Patina is not only alkaline copper chloride, but basically it is a compound in different proportions of carbonate and copper (II) hydroxide and having a color from dark green (which is clearly visible in Figure 17) to blue and corresponding to the structure of the minerals malachite and azurite, respectively. It is recommended, where permissible, to use the well-accepted term Patina in the manuscript.

Point 10: Figures 1, 2, 3, 4, 10, 11, 15 and 16. Increase the font size both inside the figures and on the scales.

Response 10: We sincerely thank the reviewer for careful reading. As suggested by reviewer, we have corrected the Figures.

Comments on the response. This has not been completely corrected.

Point 11: Fig. 7a. Can be replaced for better understanding (in comparison with Figure 7b) with an even smaller magnification (for example, 2000 x).

Response 11: We sincerely thank the reviewer for careful reading. I don't think smaller multiples are necessary. SEM images of cross-sections of LN/PAANa/GC gel-coated rusted copper copper plates were taken to study the internal structure of the gel. A multiplier of 5000 is more than enough for analysis.

Comments on the response. The remark concerned the fact that for greater persuasiveness it is recommended to make the difference in magnifications between the drawings more than twice (as it is now), for example, four times.

Point 15: Figure 11a. Reduce the number of values along the Y axis. What is depicted on the insert is not visible at all absolutely. The insertion is not mentioned in the text.

Response 15: We think this is an excellent suggestion. I have reduced the number of values along the Y axis. The insertion is “bare bronze flake”. Because it's too small of an arc to be easily discerned, I use the insertion part to discern. I have presented the revised image in the manuscript.

Comments on the response. The remark has not been completely corrected. In addition, after removing the insert, you need to change the corresponding figure.

Point 18: Fig. 17. There is no indication (not in the text, nor in the caption to the figure) of the method by which these photos were obtained. Is it an optical microscope? Then there are no instructions in the experimental part either. There is no dimension scale.

Response 18: Thank you for your question. Fig.17 shows the photographs taken by the camera. I have presented the revised text in the manuscript. It is highlighted by blue in the revised version for easier tracking.

Comments on the response. It's good for readers to also find out that these are optical images, and it would also be nice to put a scale (as previously recommended to the authors, but it has not been corrected so far) or some dimensional object on the picture so that the sizes of the samples are estimated.

Point 19: References of literature here are not designed according to the rules of MDPI. DOI indexes are missing everywhere. What is the "web" at the end of each refs? Some of the authors' names are given with distortions (especially hit the authors of ref.1 got: Correct please “K. Marušić, H. Otmačić-Ćurković, Š. Horvat-Kurbegović, H. Takenouti, E. Stupnišek-Lisac ….).

Response 19: We feel sorry for our carelessness. In our resubmitted Manuscript, the typo is revised. Thanks for your correction.

Comments on the response. The remark is almost not corrected. The list of references is full of new typo. Oddly enough, the design has become even worse compared to the original version! What are the author names in the new refs 32,33? In addition, the authors did not highlight the changes made in the new list of references.

In the end, unfortunately, we can say that now almost everything is designed very carelessly, even the formulas do not use subscript characters, which is completely impossible to understand, for the subject of chemistry is this manuscript? The authors probably have no experience of publishing in international journals. In this case, it can be recommended to include  a new co–author (among your colleagues): an experienced specialist and who would help correct and arrange everything correctly.

Author Response

Reviewer 2 Report (Previous Reviewer 1)

The present manuscript has been consderably ameliorated, as compared to the previous version. It can be accepted for publication.

A minor comment:

Scheme 2.  What is the meaning of presenting the chloride anions being separated  from sodium cations? The authors try to express a charge separation in this size scale?

Also, check lines 401-405. A more correct writting is necessary.

Dear Editor,

I think that English is quite correct. Some minor checks should, of course, be done (see, for example, my comment to the authors).

Best Regards

Georgios Bokias

Round 2

Reviewer 1 Report (Previous Reviewer 2)

For schemes

“This multicolored object is intended to represent the form of a coated bronze plate and to differentiate it from scheme 2, which has a change in mechanism after aging. Its internal structure has been indicated by the arrows in the figure below.”

(1) Perhaps then it is necessary to write what it is in the scheme opposite the object. (2) A mechanism is a sequence of process stages. In the scheme 1 it is. In the scheme 2 – no, but it is signed that "mechanism", which is unclear. That is why the idea of combining 1 and 2 into a common, possibly more understandable scheme arises. (3) Design note: The added fourth stage contains many small (poorly distinguishable) objects and even the arrow is different (carelessly). It is advisable to increase both the objects and the font, and it is better to increase this entire stage, due to the importance, entirely. The same is poorly discernible in Figure 1b.

 For references

The design of the literature still leaves much to be desired. Ref 16 (even if the DOI does not exist) is it a book, an article, a review, a report? It's not the -  doi:10.1002/pol.1965.100030720 or doi:10.1002/pol.1965.100030721 ?

Ref 2, the beginning of the title is what language is used?

Author Response

This manuscript is a resubmission of an earlier submission. The following is a list of the peer review reports and author responses from that submission.

Round 1

Reviewer 1 Report

The present work presents an interesting attempt to prepare self-healing composite hydrogels for anticorrosion coating purposes.

The manuscript can be accepted for publication, after considering the following remarks:

i.                 A clear presentation of the chemical structure of the composite hydrogel would be very helpful. The one used in Figure 1 might be good. As inset in Figure 1, it is rather invisible. I would suggest giving a large visible version of this structure as a second part of Scheme 1.

ii.                Scheme 1. PAANa is a polyelectrolyte, just like GC. I wonder whether it is correct to represent PAANa as a small red sphere and not as an elongated chain. (see, also next comment om molecular weights).

iii.               Section 2.1 Please give details on the polymers used, if available (molecular weight for both polymers; deacetylation degree and hydroxyethylation degree for chitosan). Also, please give details on purity, concentration etc for the rest chemicals used. The lines 97-106 from this subsection should be removed. The information for each instrument can be given in section 2.7.

iv.               Thermogravimetric analysis. The authors try to discuss these results on the basis of interactions between different components. To be correct, the TGA curves of pure components should be given.

v.                Section 3.6. What is the behavior of LN/PAANa, as well as GC coatings?

vi.               The authors should decide to use a single term for the chitosan sample. In fact, we find four terms, now: chitosan, hydroxyethyl chitosan and ethylene glycol chitosan and

vii.             The manuscript should be checked for typos and, in some cases, the use of English language.

viii.            The quality of Figures in many cases is not adequate.

English is quite good. However. it should be checked during proof-editing for minor problems.

Reviewer 2 Report

The composition of a patina, and the corrosive process leading to changes in copper and bronze, are different when objects are buried in soil. Corrosion of archeological metallic objects is very much dependent on the acidity of the soil and the type and amount of minerals and organic material lying in the vicinity of the objects. The purpose of article is actual. The paper considers the use of supramolecular gel in combination with chitosan (this is the main innovation in the work) to obtain more effective coatings that inhibit the corrosive effects of the external environment for bronze samples. The successful results of the authors' work have an important practical application in the preservation of cultural heritage. All comments on this work are minor, although there are quite a lot of them and they should be corrected.

1.     The introduction should be redone. The introduction should be shortened, the purpose of the work should be clearly stated and remove your results. The scheme may be presented in the discussion of the paper or in the conclusions. It is reasonable to combine schemes 1 and 2.

2.     Schemes 1 and 2 can hardly be called "mechanisms", the mechanism requires bringing stages. This is most likely the principle. At the same time, great respect to the authors for the fact that they bring schemes. Unfortunately, in their present form they are difficult to understand. A more detailed decoding of everything with arrows that is present on them is required. You should also increase the font size.

3.     It is recommended to make a table with abbreviations and formulas of the substances used and their combinations.

4.     FTIR spectra, TG and DTG curves reasonable represent in supplementary materials.

5.     Line 95. What is formula of “sodium chloride acetone”?

6.     Section 2.1. Not all countries-manufacturers of this equipment are indicated in the description of the devices.

7.     Section 2.7. It is recommended to break at least into paragraphs according to the description of each method.

8.     What is the chemical composition investigated “bare” bronze and “rusted” bronze?

9.     Why authors used term “rusted bronze” and no “patina bronze” (https://doi.org/10.1016/j.culher.2021.09.007)? The term “rusted” often concern of iron.

10.                       Figures 1, 2, 3, 4, 10, 11, 15 and 16. Increase the font size both inside the figures and on the scales.

11.                       Fig. 7a. Can be replaced for better understanding (in comparison with Figure 7b) with an even smaller magnification (for example, 2000 x).

12.                       Line 308-309. “As shown in Fig. 7a, the thickness of the gel coating was ~2 μm.”: It would be even better to indicate this thickness with arrows directly in Fig. 7.

13.                       The equation-calculation on Lines 273-274 is better to display on a separate line.

14.                       Table 1. What is the meaning of the index or reference "1" after the value "-9.848".

15.                       Figure 11a. Reduce the number of values along the Y axis. What is depicted on the insert is not visible at all absolutely. The insertion is not mentioned in the text.

16.                       For future research of the authors: polarization curves of corrosion would undoubtedly decorate the work and would unequivocally confirm the success of the proposed procedures.

17.                       Figure 16. The caption to the figure does not contain an indication of the method and analysis used.

18.                       Fig. 17. There is no indication (not in the text, nor in the caption to the figure) of the method by which these photos were obtained. Is it an optical microscope? Then there are no instructions in the experimental part either. There is no dimension scale.

19.                       References of literature here are not designed according to the rules of MDPI. DOI indexes are missing everywhere. What is the "web" at the end of each refs? Some of the authors' names are given with distortions (especially hit the authors of ref.1 got: Correct please “K. Marušić, H. Otmačić-Ćurković, Š. Horvat-Kurbegović, H. Takenouti, E. Stupnišek-Lisac ….).
